# A Comparison of Inertial Measurement Units and Overnight Videography to Assess Sleep Biomechanics

**DOI:** 10.3390/bioengineering10040408

**Published:** 2023-03-25

**Authors:** Nicholas Buckley, Paul Davey, Lynn Jensen, Kevin Baptist, Angela Jacques, Bas Jansen, Amity Campbell, Jenny Downs

**Affiliations:** 1Curtin School of Allied Health, Curtin University, Perth 6102, Australia; 2Telethon Kids Institute, Perth 6009, Australia; 3Physiotherapy Department, Perth Children’s Hospital, Perth 6009, Australia; 4Ace Therapy Services, Perth 6021, Australia; 5Institute for Health Research, University of Notre Dame Australia, Fremantle 6160, Australia

**Keywords:** overnight videography, XSENS DOTs, sleep biomechanics

## Abstract

Purpose: The assessment of sleep biomechanics (comprising movement and position during sleep) is of interest in a wide variety of clinical and research settings. However, there is no standard method by which sleep biomechanics are measured. This study aimed to (1) compare the intra- and inter-rater reliability of the current clinical standard, manually coded overnight videography, and (2) compare sleep position recorded using overnight videography to sleep position recorded using the XSENS DOT wearable sensor platform. Methods: Ten healthy adult volunteers slept for one night with XSENS DOT units in situ (on their chest, pelvis, and left and right thighs), with three infrared video cameras recording concurrently. Ten clips per participant were edited from the video. Sleeping position in each clip was coded by six experienced allied health professionals using the novel Body Orientation During Sleep (BODS) Framework, comprising 12 sections in a 360-degree circle. Intra-rater reliability was assessed by calculating the differences between the BODS ratings from repeated clips and the percentage who were rated with a maximum of one section of the XSENS DOT value; the same methodology was used to establish the level of agreement between the XSENS DOT and allied health professional ratings of overnight videography. Bennett’s S-Score was used to assess inter-rater reliability. Results: High intra-rater reliability (90% of ratings with maximum difference of one section) and moderate inter-rater reliability (Bennett’s S-Score 0.466 to 0.632) were demonstrated in the BODS ratings. The raters demonstrated high levels of agreement overall with the XSENS DOT platform, with 90% of ratings from allied health raters being within the range of at least 1 section of the BODS (as compared to the corresponding XSENS DOT produced rating). Conclusions: The current clinical standard for assessing sleep biomechanics, manually rated overnight videography (as rated using the BODS Framework) demonstrated acceptable intra- and inter-rater reliability. Further, the XSENS DOT platform demonstrated satisfactory levels of agreement as compared to the current clinical standard, providing confidence for its use in future studies of sleep biomechanics.

## 1. Introduction

The assessment of movement and position during sleep (collectively, sleep biomechanics) is of clinical utility in a wide variety of conditions, including Obstructive Sleep Apnoea [1] and Parkinson’s Disease [2]. Assessment of sleep biomechanics is of particular relevance to those with neurodisability, as insufficient overnight repositioning has long been suspected as contributing to the development of body shape distortion. First proposed by Fulford and Brown [3], the theory posits that those with severe physical disability (such as individuals with cerebral palsy, Gross Motor Function Classification System [4] levels IV and V) are less able to vary their position throughout the night through movement (typically by rolling over), and so spend more time in fewer positions compared to their typically developing peers [5]. This lack of positional variation results in uneven exposure to gravity, causing bone and connective tissues to be laid down asymmetrically [6]. Resulting body shape distortions (primarily presenting in cerebral palsy as scoliosis, pelvic obliquity, hip migration and chest wall deformity [7]) have negative health impacts, including pain, [8] impairment of respiratory function [9], sleep disturbance [10] and overall loss of function [11]. However, confirmatory evidence of this proposed difference in sleep biomechanics and the inability to reposition between people with severe physical disability and the general population is lacking. There are two main reasons: (1) an absence of an accepted data capture tool, and (2) no standard format to report sleep biomechanics.

There is no commonly accepted methodology for capturing sleep biomechanics data. Particular attention has been paid to sleep position; studies that have examined sleep position are methodologically diverse, with a variety of methods such as videography [12], pressure sensors [13] and wearable accelerometers [14] being used, rendering a synthesis of findings difficult. The current standard clinical assessment of sleep position is overnight infrared videography (typically, recorded video footage from a single camera angle), that is then visually inspected and coded by a clinician (typically, as supine/prone/left side lying/right side lying [15]). For example, Cary et al. investigated the relationship between sleep postures and lower back pain in adult participants, utilizing infrared cameras [16]. While this method provides valuable information [17] there are several limitations; data analysis is time consuming, relies on the availability of a skilled clinician, is constrained by the physical sleep environment (lighting, bedroom dimensions) and is inherently subjective, so is vulnerable to rater error. Further, there is no standard methodology of processing, analysing and presenting sleep biomechanical data once it is captured. Comparison of studies in this area is vexed by a lack of consistency in defining what constitutes a ‘repositioning’ event. For example, both 45 degree trunk rotation [5] and the movement of at least three limbs [18] have been proposed as definitions of ‘repositioning’. Other studies have proposed more complex analysis methods; Wrzus et al., for example, utilized a decision tree paired with an angle-based segmentation algorithm [19]. While methodologically sound this is complex and so may render its implementation as a clinical tool (for use and interpretation by clinicians, patients and families) difficult. Additionally, many of these methods only include axial trunk rotation and do not capture limb movement, limiting understanding of how body segments may influence each other’s position during sleep.

There is a need for an easily implemented, objective measure of sleep biomechanics that can be used as the standard assessment tool in research and clinical practice—a need currently unmet by the available methods. A set of wearable Inertial Measurement Units (IMUs) is putatively more efficient, has a lower skill requirement and is more objective and replicable than clinician-rated overnight videography. These devices have been used to good effect previously for the measurement of sleep position in children [14] and adults [19]. In addition, there is a need for a novel classification system for sleep biomechanics that (a) can be applied broadly across diverse research and clinical applications, (b) comprehensively describes the orientation of multiple body segments in relation to each other, and (c) has a streamlined, automated analysis process, providing results that are easily interpreted by clinicians, patients and their families—such an analysis process does not currently exist.

This study aimed to validate the use of a novel wearable sensor system (XSENS DOTs), paired with a new classification of position during sleep, the Body Orientation During Sleep (BODS) Framework. We aimed to answer the following questions:

What is the intra-rater reliability of sleep position measured using overnight infrared videography?

What is the inter-rater reliability of sleep position measured using overnight infrared videography?

What is the level of agreement between clinician-rated overnight infrared videography and a novel automated system (XSENS DOT orientation data, coded through the BODS Framework) in assessment of sleep position?

## 2. Methods

### 2.1. Working Definitions

The working definitions for the terms used in this paper are shown in the Box 1.

Box 1A list of working definitions for terms used in this paper.Sleep Biomechanics: Collective term for
movements made and positions adopted during sleep. (Body) Segment: Referring to a given
anatomical region of the body—for this study, particularly referring to the
attachment areas for the DOTs: sternum (anterior chest), pelvis (anterior
abdomen) and left and right thighs (immediately superior to the patella).DOTs: Small, multipurpose IMUs from XSENS
(Enschede, Netherlands), programmed using a smartphone with the native DOT
app installed.BODS: The Body Orientation During Sleep
Framework. A novel classification system for sleep biomechanics, splitting
the 360° of orientation in the transverse plane into component sections to describe
lying position (lying being the usual sleeping position). (BODS) Section: One of the component sections
making up the BODS. In the embodiment of the BODS used for this study, there
are twelve sections in total. Each cardinal position (supine/prone/left side
lying/right side lying) is 5° in size, and the other 8 sections are 42.5° in
size.(Sleep) Position: As defined by the BODS, the
orientation of the given body segment in the transverse plane at a given
point in time. (Sleep) Movement: As defined by the BODS, a
change in orientation of sufficient amplitude that it results in a change of one
of more BODS sections. Cardinal Positions: Collective term for
sections 12 (supine), 3 (left side lying), 6 (prone) and 9 (right side
lying).Rater: An experienced clinician, reviewing
captured video footage to assess and code the sleep position according to the
BODS.Difference Score: When comparing two data
capture systems’ outputs (in this case, the DOTs and human-rated overnight
videography) and the sum of the BODS sections, the two data sets differ by
when classifying the same sleeping position using the BODS Framework. 

### 2.2. Ethics

This study received approval from Human Research Ethics Committee at Curtin University (HRE2020-0138).

#### Participants

Ten healthy volunteer adults (18–60 years old) were recruited via convenience sampling for monitoring of sleep positions. Adults were recruited to optimise compliance with specific instructions on the study procedures and provide consent for overnight video. Exclusion criteria included the presence of acute or chronic injuries or pain that would have limited mobility when lying, as well as allergy/irritation from adhesive dressings.

Raters were approached by the lead author to participate in the study. All those asked to participate provided oral consent with the understanding that this was voluntary, and they could change their mind at any time. Three occupational therapists and three physiotherapists were recruited to rate the captured video footage. All clinicians were experienced in the neurodisability field with >15 years of clinical experience and were all skilled in clinical postural assessments of people in the lying position, typically for the prescription of sleep support equipment.

### 2.3. Assessment Instruments

#### 2.3.1. XSENS DOT Platform

The XSENS DOTs (XENS, Enschede, The Netherlands) (Version 2.0, 4 Hz sample, ±2000°/s gyroscope, ±16 g accelerometer, ±8 Gauss magnetometer) are a set of small IMUs (36 × 30 × 11 mm, 10.8 g). XSENS DOTs (abbreviated to ‘DOTs’) were placed on the sternum, pelvis, left thigh and right thigh (one sensor per body segment, see Figure 1 for placement locations) and were used to collect the orientation of each of those body segments. The DOTs were programmed by and logged data via Bluetooth to a Samsung Galaxy S9 Smartphone (Samsung, Seoul, Republic of Korea) using the native XSENS DOT application (v.1.6, XENS, Enschede, The Netherlands). The DOTs were secured using 100 mm × 100 mm Tegaderm (3M, Saint Paul, MA, USA) transparent film dressing over the top of each DOT.

#### 2.3.2. Body Orientation during Sleep Framework (BODS)

The Body Orientation During Sleep Framework (BODS) is a novel method of classifying body position developed specifically for the quantification of sleep biomechanics (Figure 2). The BODS divides the 360° of possible transverse rotation (in lying) into 12 sections; division into 12 sections was initially chosen for ease of visual reference (similar to a clock face). When a person is lying down the DOT orientation will always be in one of the sections. For example, the output of a DOT attached to the sternum of a participant lying in supine would be coded as being in Section 12. When the participant is lying in prone, the output of the DOT would be coded as being in Section 6 (Figure 2).

The cardinal positions of supine (Section 12), prone (Section 6), left side lying (Section 3) and right side lying (Section 9) were given smaller 5-degree sections to document symmetrical lying. The sections in between the cardinal positions were divided into two, comprising 42.5° each. The BODS is classified from the perspective of the rater being at the foot of the bed.

#### 2.3.3. Video Cameras

Overnight video footage was captured by three FDRAX33 digital video cameras (SONY, Tokyo, Japan; 4K Ultra HD (3840 × 2160), 29.8 mm ZEISS Vario-Sonnar T* lens, Oberkochen, Germany), using the NightShot infrared recording mode. The video cameras were positioned at the foot of the bed and laterally (left and right of the bed) on tripods (see Figure 3). The camera at the foot of the bed was elevated in order to provide a full body longitudinal view.

### 2.4. Data Collection Procedure

Data collection of overnight videography occurred in the participant’s native sleeping environment; all participants slept in their usual bed and bedroom. The investigator attended each participant’s home and set up the camera array and an Infrared Illuminator (Securview, Edison, NJ, USA) to provide an additional infrared light source to ensure the captured footage was of high enough resolution. The investigator provided each participant with a short training session on how to attach and start recording with the DOTs using the provided smartphone and XSENS DOT app, as well as written instructions (see Appendix A). As the entire body needed to be visualised, participants slept without bedclothes but wore usual sleeping attire. A heater (De’Longhi, Treviso, Italy) was provided to ensure thermal comfort. When participants were ready to sleep for the night, they applied the DOTs, commenced DOT recording via the smartphone app and started recording on the cameras. They then slept, ensuring they took the smartphone with them if they left the bedroom to ensure continuous Bluetooth streaming. Upon waking, participants stopped recording on the DOTs and the cameras. The following morning the investigator attended the participant’s home and removed all study equipment.

### 2.5. Data Processing

#### 2.5.1. Camera Footage

Camera footage was downloaded and edited into a standardised layout using Premiere Pro video editing package (Adobe, San Jose, CA, USA) (Figure 4). Using a custom Labview program (National Instruments, Austin, TX, USA) repositioning throughout the night were identified by changes in DOT orientation, and time-matched clips of approximately 30 s movement duration (with 10 s buffer of footage prior to and after movement) were extracted from the video footage. For each participant, a 10-clip reel was prepared, consisting of six repositionings, two clips of the participant lying in a stationary position, one repeated repositioning clip, and one repeated stationary clip. The order of these clips was randomised for each clip reel. One hundred clips in total were extracted (10 clip reel for each of the 10 participants).

#### 2.5.2. Rating of Camera Footage

Prior to rating, each rater was provided with a single 60 min training session. This consisted of a pair of training videos on how to rate according to the BODS framework (see Appendix A) and three practice clips to rate. Following this session, raters were provided with the clip reel for each of the 10 participants. Raters provided a rating for each body segment (chest, pelvis, left and right thighs) independently, reporting the BODS section for each body segment at both the beginning and the end of the clip. To ensure consistency between raters, only body segment position in the first and last frame of the video was assessed. Raters entered BODS data before playing the clip, then observed the footage of the movement (if any) that occurred, and then at the concluding frame of the clip, entered the BODS data for the body segment positions in that still image. Data were entered into a standard form (see Appendix A).

#### 2.5.3. XSENS Data

After being downloaded from the smartphone, quaternion recordings were converted to Euler angles using a ZXY decomposition. The quaternion method of representing sensor orientation was selected during recording due to smaller storage size of files and convenience of data handling; however, for ease of clinical interpretation, data were ultimately output to the Euler angle format commonly used in biomechanics. Each sensors’ pitch angle (corresponding to rotation in the transverse plane for the given segment) was then coded using the BODS framework, giving each body segment an orientation from 1–12 (see Figure 3). The BODS section for each body segment (chest, pelvis, L and R thighs) was identified at the beginning and end of each clip.

### 2.6. Statistical Analysis

#### 2.6.1. Intra-Rater Reliability

To examine intra-rater reliability for each human rater, difference scores were calculated for the two ratings provided by each rater for each of the repeated clips, represented as the maximum difference in number of sections between the two ratings. For example, if the first BODS rating was Section 5, and the second was Section 7, this would be a difference score of 2 (the difference in BODS sections). The maximum difference score possible was 6 (e.g., a DOT classification of 12 and a rater classification of 6). A difference score was calculated for each of the four dots in each repeated video clip (sternum/pelvis/left leg/right leg). Proportions of the differences (1–6) were calculated to determine highest percentage accuracy within the raters.

#### 2.6.2. Inter-Rater Reliability

To examine agreement between human raters, Cohen’s Kappa and Bennett et al.’s S-Score were used to compare BODS classifications between the six raters for each of the 100 clips. Bennet’s score was included in addition to the standard Cohen’s Kappa due to the potential for small differences between raters resulting in very low Kappa values, the high agreement/low Kappa paradox [20]. Cohen’s Kappa could result in an underestimation of inter-rater reliability if used in isolation; Bennett’s S score was therefore included as this measure is better able to handle smaller marginal differences.

#### 2.6.3. Raters vs. DOTS

To compare the level of agreement between the raters and the DOTs, difference scores (how many BODS sections two ratings differ by) were calculated for the corresponding DOT and human ratings for each clip. For example, a DOT classification of 12 and a rater classification of 12 would give a difference score of 0; a DOT classification of 12 and a rater classification of 11 would give a difference score of 1. A difference score was calculated for each of the four dots in each video clip (sternum/pelvis/left leg/right leg).

## 3. Results

### 3.1. Participant Characteristics

All 10 participants completed data collection and no difficulties or adverse events were reported. There were seven males, and age ranged from 20 to 59 years (mean 29.4 years, median 27.5 years).

#### 3.1.1. Intra-Rater Reliability

All six raters demonstrated a high level of intra-rater reliability, with >48% of ratings with an error score of 0 and >90% of ratings with an error score of ≤1, for all body segments (Table 1). There was slightly higher agreement for chest, pelvis and left leg ratings (mean difference score of ≤1 segment across all raters of 97%, 99%, 98%, respectively) compared with right leg ratings (93%).

#### 3.1.2. Inter-Rater Reliability

Calculation of Cohen’s kappa for comparisons between each rater resulted in a range from 0.254 to 0.470, indicating a low to fair level of agreement. Calculation of Bennett et al.’s S-score [20] gave a range of 0.466 to 0.632, indicating moderate agreement between raters (Table 2).

#### 3.1.3. Raters vs. DOTS

Overall, the human raters demonstrated a high level of agreement with the DOTs, with >81% of ratings with a maximum difference of 1 section (Table 3). The chest and pelvis ratings displayed a higher level of agreement (mean difference score of 0 segment difference 38% and 53%, and maximum difference of 1 segment of 97% and 96%, respectively, across all raters) when compared to leg ratings. The left and right leg ratings were considerably lower (mean difference score of 0 segment difference 28% and 33%, and maximum difference of 1 segment of 88% and 89%, respectively, across all raters). The raters reported occasional difficulty with confusing left and right legs during rating; visual inspection of clips where there were particularly high difference scores (5 and 6) confirmed that this was the most common cause of rater error.

## 4. Discussion

This study first aimed to establish the intra- and inter-rater reliability of the current standard clinical tool for assessing sleep positioning, overnight videography rated by clinicians, as codified through the novel BODS framework. Based on the results of this study, we found a high level of intra- rater reliability and moderate level of inter-rater reliability. This study also found high levels of agreement on body position measured by clinician rated overnight videography and XSENS DOTs.

The relatively high levels of agreement demonstrated between and within human raters is encouraging, as it indicates that the existing clinical standard of overnight videography is reliable and so it is appropriate to use it as a benchmark with which to compare new assessments, such as the XSENS DOTs. The good agreement lends further credence to previous studies that have utilised overnight videography to assess sleep position [16,18,21]. One reason may be due to the high level of clinical skill in the human raters, all of whom have 15+ years of experience working in the disability field, routinely performing postural assessments of individuals with neuroimpairments. This specialised experience is not always available in clinical settings, and so use of overnight videography by less experienced clinicians may result in less consistent results. An objective tool such as the XSENS DOTs is likely superior in this respect as it does not rely on the clinical skill of its user and can be easily implemented to accurately measure sleep biomechanics regardless of the user’s breadth of experience.

In general, the accuracy of the human raters compared well to that of the XSENS DOTs. While a high proportion of human ratings were a maximum of 1 section different to the corresponding DOT rating, the majority of the sections are large (42.5°) and so variability at a granular level could be greater. Narrower (5°) sections could be used for the cardinal positions, as in these positions the body tends to be more symmetrical and so the position of segments more easily rated. The wider range for the non-cardinal sections of the BODS were necessary due to the possible visual limitations of human raters. Smaller non-cardinal sections would have offered more specific data, but were deemed unrealistically small for even skilled, experienced raters to be able to distinguish. These limitations of human raters have been evidenced in other studies of visually assessed segment position. For example, Abbot et al. demonstrated in a group of physiotherapists that the smallest visually detectable change in range of various dynamic joint angles (lumbar flexion, knee flexion, shoulder abduction) was 9° to 12° [22]. This theory was further borne out by some of the difficulties experienced by the human raters in our study, even with the wider sections of the BODS. As seen in Table 3, there was less agreement for left leg/right leg positions compared to the sternum and pelvis positions, likely due to the greater range of movement at the hip (compared to the spine, which is primarily responsible for sternum and pelvis positioning) resulting in a wider variety of positions being possible for the thigh segments, as compared to the potential positions of the sternum and pelvis. Further, raters reported occasional confusion when distinguishing between left and right, which may account for some of the 5 and 6 difference scores that were present.

Despite being previously unfamiliar with the BODS, following the brief training session raters were able to effectively use the BODS framework to rate body orientation during sleep. The use of this system demonstrates a unique advantage over other classification systems, as the orientation of each body segment can be described, and a composite of each segment describes the entire body, a capacity absent from other systems. For example, Brown et al. assessed only gross body position and coded only in cardinal positions (supine/prone/left side lying/right side lying) [23]. Further, because the DOTs performed well against both a lab-based standard reference system (VICON motion analysis) [24] and the current clinical standard of overnight videography, future studies utilising the DOTs alone need not be constrained by human perception, and different versions of the BODS with finer gradations can be used. While this study focused primarily on classification of static positions, the BODS (with data captured by the DOTs) also allows for recording the number of repositionings/night by registering the number of transitions between sections. The BODS/DOTs combination also allows for exposure analysis of the duration participants rest in each section. These outcome measures can be used in future research to investigate sleep biomechanical phenotypes for clinical populations (e.g., cerebral palsy) and contrast this with the sleep biomechanics of the general population. Moreover, when paired with an objective, computerised system such as the XSENS DOTs the BODS allows for streamlined, automated data analysis and simplified interpretation of findings, putatively lowering the threshold for adoption into clinical practice and research methodology. In the field of neurodisability, simplified assessment of sleep biomechanics would enable easier and more effective prescription of supported lying equipment, improving sleep comfort and quality of life for people with neurodisability [25].

There are several strengths in this study. Raters were highly experienced allied health clinicians from a variety of clinical backgrounds. The study design captured full variability of sleep biomechanics, testing accuracy of classification in a variety of situations including moving, stationary and repeated clips. This study leveraged a large data set relative to its modest sample size (*n* = 10) by extracting a wide variety of clips (100 per participant) and rating each of the four body segments’ positions at the beginning and end of each clip, resulting in 800 individual ratings (100 clips × 2 ratings per clip × 4 body segments) from each of our six raters, resulting in 4800 total ratings. We are confident that this provided an adequate data set with which to compare the DOTs to the clinical standard of overnight videography. This was accompanied by a relatively simple and easily interpreted metric of percentage agreement between the two systems, which clearly demonstrated the high level of intra- rater reliability and high levels of agreement on body position measured by clinician-rated overnight videography and XSENS DOTs.

A limitation of this study was that head orientation was not recorded, as it has been in other studies [14]. The investigators chose not to measure head orientation in this study as trunk and leg position was the primary focus, and out of concern for comfort and unduly influencing naturalistic sleep biomechanics [26]. The inclusion of other XSENS DOTs (e.g., on the forehead to measure head orientation, or on the upper limbs) and the utility of these additional data to the holistic assessment of sleep biomechanics could be a feature of future study designs. Additionally, as previously discussed, the BODS sections in between the cardinal position sections were wide (42.5°), which proved to be a barrier to more granular comparisons of findings between the systems. While this was deemed necessary to allow for visual assessment by human raters, future studies utilising DOTs-captured, BODS-rated data to assess sleep biomechanics could use smaller section sizes (e.g., 5° or 10°) to allow for more detailed description of sleep position and movement. In addition, future studies could use different configurations of the BODS (i.e., a greater or lesser number of sections) depending on the research questions and purpose of analysis.

Compared to current systems, the XSENS/BODS system has a number of distinct advantages. It is easily implemented in the ecological sleeping environment of each participant; this is not always feasible with systems that require large, cumbersome equipment, such as overnight videography (e.g., as proposed by Cary et al. [16]). Further, the data analysis process is far simpler using the BODS; this stands in contrast to the onerous process of manually reviewing hours of video footage, or else using a complex system of classification and analysis as is used in current assessments that utilized accelerometry or IMUs. Further, we are confident that the system presented in this study will yield results that are more easily integrated into current clinical practice due to the superior usability and ease of interpretation of findings (by families and clinicians) as opposed to current solutions (for example, Wrzus et al. [19]).

## 5. Conclusions

Based on the results of this study, we recommend the use of the XSENS DOTs in future studies examining sleep biomechanics—they are more accurate than human raters and allow for more detailed analysis of sleep biomechanics in a far less labour-intensive manner. Future studies could focus on further development of the BODS framework and trialling of both it and the DOTs in diverse clinical populations.

## Figures and Tables

**Figure 1 bioengineering-10-00408-f001:**
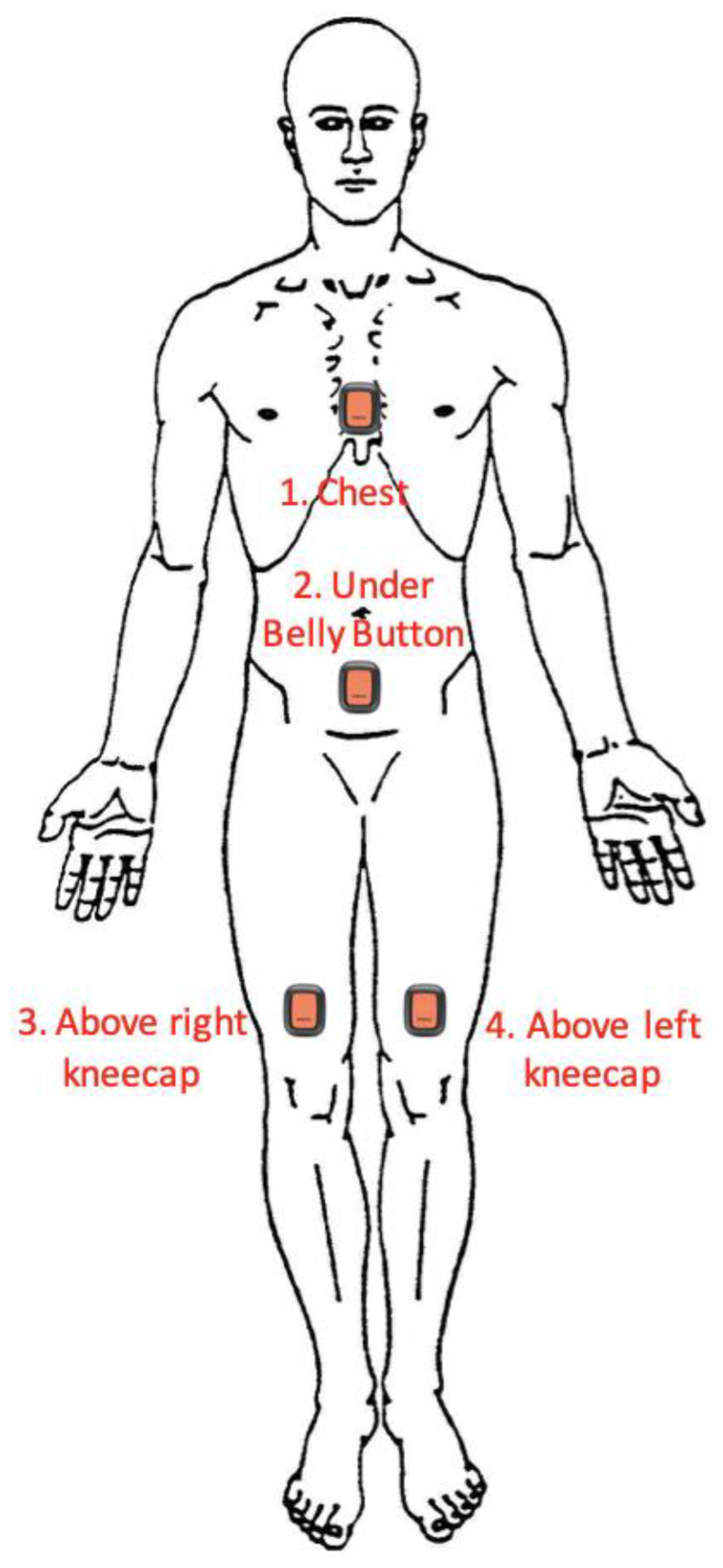
Locations of XSENS DOT Placement on Participants.

**Figure 2 bioengineering-10-00408-f002:**
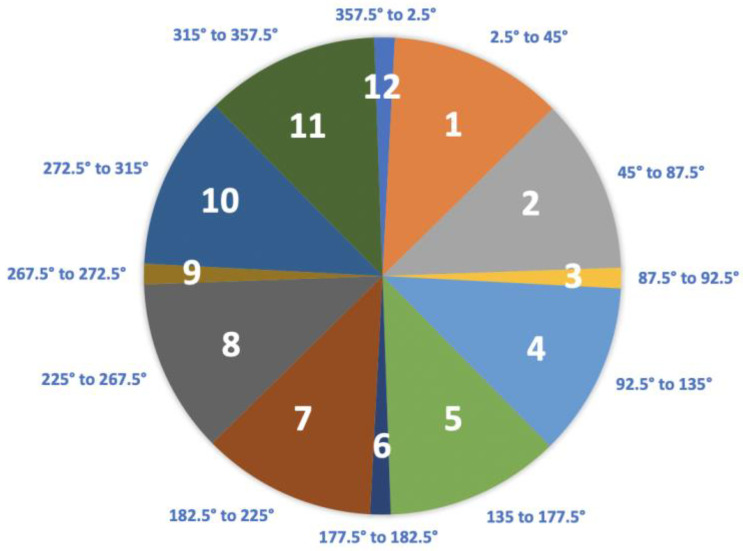
The Body Orientation During Sleep Framework (BODS).

**Figure 3 bioengineering-10-00408-f003:**
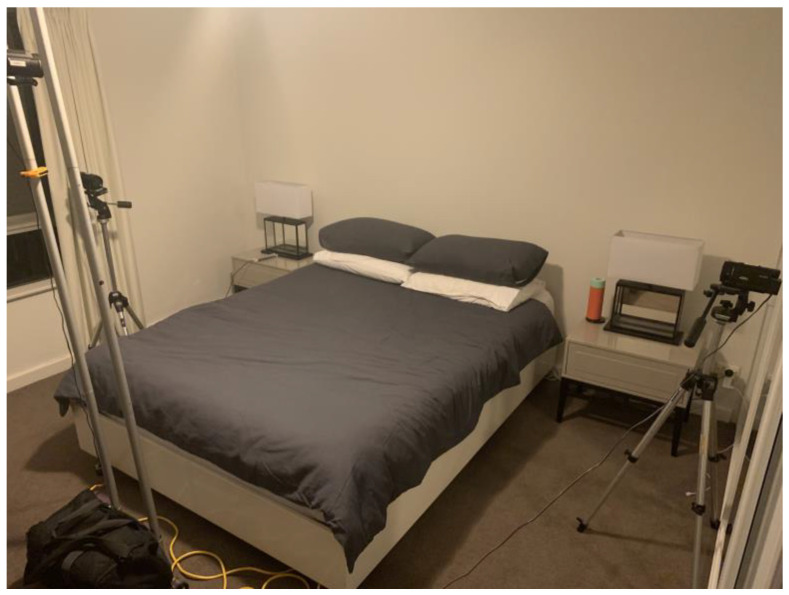
Camera Arrangement for Overnight Recording.

**Figure 4 bioengineering-10-00408-f004:**
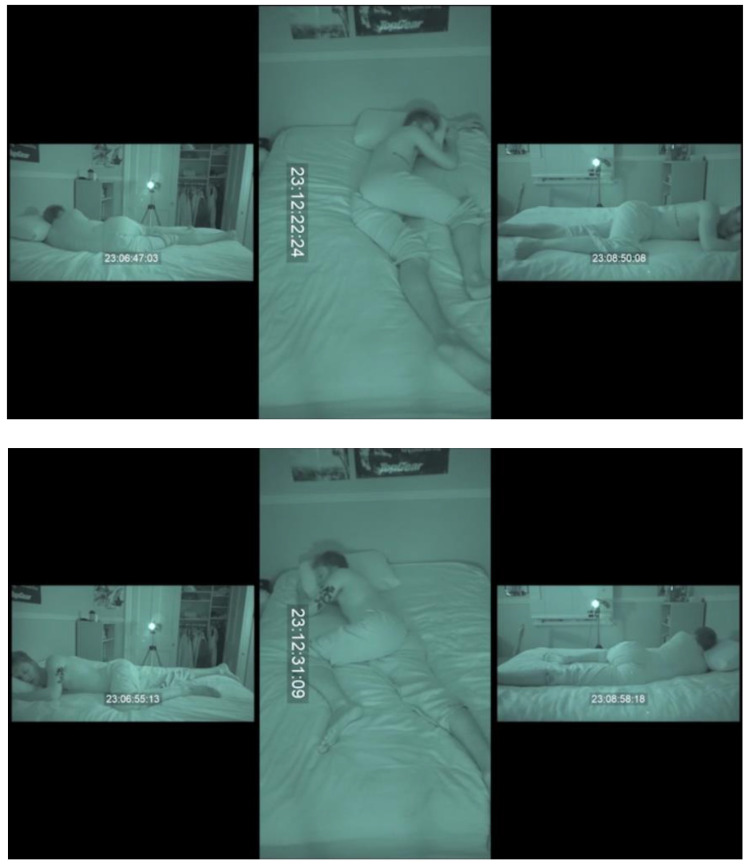
Example Frame of Clip Footage presented to human raters. * Reproduced with consent from pictured participant. Example Clip Video: https://youtu.be/SbKXkGrbrak (accessed on 4 July 2021).

**Table 1 bioengineering-10-00408-t001:** Frequency and percent of Intra-Rater Reliability of Human Raters (by Difference Score for each body segment).

		Sternum		Pelvis		Left Leg		Right Leg	
		Frequency	Percent	Frequency	Percent	Frequency	Percent	Frequency	Percent
Rater A Difference Scores	0	21	53%	26	65%	23	58%	19	48%
	1	15	38%	11	28%	15	38%	17	43%
	2	1	3%	2	5%	1	3%	2	5%
	3	1	3%	0	0%	0	0%	1	3%
	4	1	3%	0	0%	0	0%	0	0%
	5	0	0%	0	0%	0	0%	0	0%
	6	1	3%	1	3%	1	3%	1	3%
	Total	40	100%	40	100%	40	100%	40	100%
Rater B Difference Scores	0	24	60%	33	83%	26	65%	29	73%
	1	16	40%	7	18%	13	33%	9	23%
	2	0	0%	0	0%	1	3%	2	5%
	3	0	0%	0	0%	0	0%	0	0%
	4	0	0%	0	0%	0	0%	0	0%
	5	0	0%	0	0%	0	0%	0	0%
	6	0	0%	0	0%	0	0%	0	0%
	Total	40	100%	40	100%	40	100%	40	100%
Rater C Difference Scores	0	22	55%	37	93%	27	68%	28	70%
	1	17	43%	3	8%	12	30%	9	23%
	2	0	0%	0	0%	1	3%	3	8%
	3	1	3%	0	0%	0	0%	0	0%
	4	0	0%	0	0%	0	0%	0	0%
	5	0	0%	0	0%	0	0%	0	0%
	6	0	0%	0	0%	0	0%	0	0%
	Total	40	100%	40	100%	40	100%	40	100%
Rater D Difference Scores	0	27	68%	34	85%	26	65%	32	80%
	1	12	30%	6	15%	13	33%	7	18%
	2	0	0%	0	0%	1	3%	1	3%
	3	1	3%	0	0%	0	0%	0	0%
	4	0	0%	0	0%	0	0%	0	0%
	5	0	0%	0	0%	0	0%	0	0%
	6	0	0%	0	0%	0	0%	0	0%
	Total	40	100%	40	100%	40	100%	40	100%
Rater E Difference Scores	0	22	55%	37	93%	27	68%	27	68%
	1	17	43%	3	8%	13	33%	10	25%
	2	0	0%	0	0%	0	0%	3	8%
	3	1	3%	0	0%	0	0%	0	0%
	4	0	0%	0	0%	0	0%	0	0%
	5	0	0%	0	0%	0	0%	0	0%
	6	0	0%	0	0%	0	0%	0	0%
	Total	40	100%	40	100%	40	100%	40	100%
Rater F Difference Scores	0	30	75%	31	78%	27	68%	26	65%
	1	10	25%	9	23%	12	30%	11	28%
	2	0	0%	0	0%	0	0%	2	5%
	3	0	0%	0	0%	1	3%	1	3%
	4	0	0%	0	0%	0	0%	0	0%
	5	0	0%	0	0%	0	0%	0	0%
	6	0	0%	0	0%	0	0%	0	0%
	Total	40	100%	40	100%	40	100%	40	100%

**Table 2 bioengineering-10-00408-t002:** Inter-Rater Reliability of Human Raters.

**Cohen’s Kappa**						
	A	B	C	D	E	F
A		0.254	0.350	0.336	0.310	0.264
B			0.433	0.413	0.344	0.397
C				0.482	0.378	0.352
D					0.470	0.351
E						0.364
F						
**Bennett’s S Score**						
	A	B	C	D	E	F
A		0.487	0.525	0.565	0.493	0.466
B			0.576	0.606	0.509	0.563
C				0.632	0.553	0.508
D					0.622	0.539
E						0.518
F						

**Table 3 bioengineering-10-00408-t003:** Results of Human Raters as compared to DOTs (by Difference Score for each body segment).

		Sternum		Pelvis		Left Leg		Right Leg	
		Frequency	Percent	Frequency	Percent	Frequency	Percent	Frequency	Percent
Rater A Difference Scores	0	73	37%	89	45%	49	25%	62	31%
	1	122	61%	104	52%	126	63%	99	50%
	2	1	1%	4	2%	23	12%	33	17%
	3	1	1%	2	1%	1	1%	5	3%
	4	2	1%	0	0%	0	0%	0	0%
	5	1	1%	1	1%	1	1%	1	1%
	6	0	0%	0	0%	0	0%	0	0%
	Total	200	100%	200	100%	200	100%	200	100%
Rater B Difference Scores	0	89	37%	111	56%	51	26%	63	32%
	1	93	61%	81	41%	118	59%	114	57%
	2	15	1%	6	3%	26	13%	18	9%
	3	2	1%	0	0%	3	2%	3	2%
	4	1	1%	1	1%	1	1%	0	0%
	5	0	1%	0	0%	1	1%	1	1%
	6	0	0%	0	0%	0	0%	0	0%
	Missing	0	0%	1	1%	0	0%	1	1%
	Total	200	100%	200	100%	200	100%	200	100%
Rater C Difference Scores	0	61	31%	109	55%	55	28%	60	30%
	1	135	68%	86	43%	118	59%	123	62%
	2	3	2%	4	2%	24	12%	14	7%
	3	1	1%	1	1%	3	2%	3	2%
	4	0	0%	0	0%	0	0%	0	0%
	5	0	0%	0	0%	0	0%	0	0%
	6	0	0%	0	0%	0	0%	0	0%
	Total	200	100%	200	100%	200	100%	200	100%
Rater D Difference Scores	0	61	31%	104	52%	60	30%	56	28%
	1	136	68%	91	46%	121	61%	122	61%
	2	1	1%	3	2%	12	6%	18	9%
	3	0	0%	0	0%	3	2%	0	0%
	4	1	1%	1	1%	1	1%	3	2%
	5	1	1%	0	0%	3	2%	1	1%
	6	0	0%	1	1%	0	0%	0	0%
	Total	200	100%	200	100%	200	100%	200	100%
Rater E Difference Scores	0	75	38%	104	52%	51	26%	68	34%
	1	118	59%	90	45%	130	65%	113	57%
	2	6	3%	6	3%	15	8%	17	9%
	3	1	1%	0	0%	4	2%	2	1%
	4	0	0%	0	0%	0	0%	0	0%
	5	0	0%	0	0%	0	0%	0	0%
	6	0	0%	0	0%	0	0%	0	0%
	Total	200	100%	200	100%	200	100%	200	100%
Rater F Difference Scores	0	106	53%	124	62%	66	33%	88	44%
	1	83	42%	61	31%	113	57%	97	49%
	2	11	6%	15	8%	19	10%	13	7%
	3	0	0%	0	0%	1	1%	2	1%
	4	0	0%	0	0%	1	1%	0	0%
	5	0	0%	0	0%	0	0%	0	0%
	6	0	0%	0	0%	0	0%	0	0%
	Total	200	100%	200	100%	200	100%	200	100%

## Data Availability

The data presented in this study are available on request from the corresponding author. The data are not publicly available due to the participants being readily identifiable (videography).

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
