# Peer review of "A Comparison of Inertial Measurement Units and Overnight Videography to Assess Sleep Biomechanics"

_bioengineering, 2023, doi:10.3390/bioengineering10040408_

Round 1
Reviewer 1 Report (Previous Reviewer 1)
The paper present a comparison between an "automatic" inertial measurement system and a "human-based" or not fully automated videosurveillance system to assess sleep biomechanics. The study seems interesting but, although the author have made some changes, it is still a bit poorly presented and edited.
The authors have made some (minimal) changes to the manuscript, but I do not think they have addressed all the issues pointed out by the different reviewers for their previous manuscript. Also I have to say that they seem to have nor payed much attention (again) to some details (like the "duplicated" review section).
The summary section is still formated in a quite "innovative" style. Please revise your paper thoroughly.
However, since most of my concerns are about the manuscript's style and not about the methodology, I think it can be published after a thorough text editing.
Author Response
Please see the attachment.

Reviewer 2 Report (Previous Reviewer 2)
The authors have addressed all of my comments accordingly, and I have no further suggestions. The manuscript can be accepted for publication in its current form.
Author Response
Please see the attachment.

Reviewer 3 Report (Previous Reviewer 3)
The authors have made necessary improvements. I think the manuscript can be accepted for publication now.
Author Response
Please see the attachment.

This manuscript is a resubmission of an earlier submission. The following is a list of the peer review reports and author responses from that submission.
Round 1
Reviewer 1 Report
The paper present a comparison between an "automatic" inertial measurement system and a "human-based" or not fully automated videosurveillance system to assess sleep biomechanics. The study seems interesting but it is a bit poorly presented and edited.
Some terms constantly used in the paper, such as "rater" are not properly described (it can be inferred, however, that it should be a human analysing the videos or something like that). Please describe better the motivation and some of the specifics of your procedures to an audience that may not be completely familiarized with them and the terminology you use.
The editing of the paper is also quite poor. Some of the figures are much bigger than necessary. I am not sure if the references are correctly formated according to the journal 's citation style (I think they are not). There are no conclusion in the paper and the summary is presented in a quite unusual style.
"Box1" and the last (unnumbered) table are also a bit weird.
Please, thorougly review and re-edit your paper, before we can discuss further issues.
Reviewer 2 Report
In this paper, sleep biomechanics was assessed and compared using inertial measurement units and overnight videography. The authors used XSENS DOT wearable sensor platform and three FDRAX33 digital video cameras to achieve the research goals. However, the paper's contribution, if any, has not been significant. The article is well-written and organized, and well-presented. The paper needs improvement before it should be accepted as follows:
1. It is necessary to add a related work section highlighting some associated works, extracting pros and cons.
2. The contribution, novelty of the paper, and the difference between the current work and previous works should be highlighted in the introduction section.
3. Technical specifications of the XSENS DOT wearable sensor and FDRAX33 digital video cameras should be provided.
4. Why the BODS are divided into 12 sections should be justified. Could it be six sections with 60 degrees for each section?
5. Is it possible to divide BODS into 12 equal sections, each section having 30 degrees?
6. Some references are outdated, for example, refs 4, 5, 7, 11, 17, and 21.
7. The conclusion section is missing in the paper. It should be added to the paper.
8. What are the potential field or diseases that can benefit from this research? Please, Discuss that.
9. The sequence of the figures must be rearranged. Figure 1 should present first.
10. The sentence on page 3, line 93, is repeated with line 94 on the same page.
11. Figure 1 is missing. Why did the paper start with Figure 2?
Reviewer 3 Report
The study validated the reliability of the coding method with videography and the XSENS DOT wearable sensor platform. The results are convincing, and the conclusion is useful for clinical application.
The manuscript was generally well written. However, the meanings of some sentences are a little difficult to understand. If possible, the author had better polish the text a little more, so that the reader can easily understand the content.
Technical problems:
1) Figure 1 was found after figure 3;
2) There is no section of conclusion.
Reviewer 4 Report
This study aimed to propose a comparative approach for inertial measurement units and overnight videography to assess sleep biomechanics. I have the following suggestions.
- What is the novelty of this study although several studies have been reported earlier for comparative study of inertial measurement units and other approaches to assess sleep?
- Authors should describe more details of the features/variables of IMU and videography used in this study in a mathematical expression.
- Authors need to add the data structure, details of the data source, and size of the dataset.
- Do authors feel only four IMU sensors are enough for understanding sleep biomechanics? How about considering the upper extremity too?
- The discussion section needs to be included. Authors must make discussion on the advantages and drawbacks of their proposed method with other studies adding a table in the discussion section.
- Clinical explanation of important findings needs to be described in support of reference.
Reviewer 5 Report
The paper presents a study focused on analyzing the reliability of the current standard clinical tool for assessing sleep position and analyzing the level of agreement on clinician-measured body position based on videography and estimates obtained using the XSENS DOT wearable sensor tool.
The study is well planned and carried out (it is methodologically correct). The manuscript is clear and concise.
The paper is of sufficient quality for publication.
